# Salt-Induced Early Changes in Photosynthesis Activity Caused by Root-to-Shoot Signaling in Potato

**DOI:** 10.3390/ijms25021229

**Published:** 2024-01-19

**Authors:** Anna Pecherina, Anastasia Dimitrieva, Maxim Mudrilov, Maria Ladeynova, Daria Zanegina, Anna Brilkina, Vladimir Vodeneev

**Affiliations:** 1Department of Biophysics, National Research Lobachevsky State University of Nizhny Novgorod, 23 Gagarin Avenue, 603022 Nizhny Novgorod, Russia; pechyorinaa@gmail.com (A.P.); ana.nedmitriewa@gmail.com (A.D.); mtengri@yandex.ru (M.M.); ladeynova.m@yandex.ru (M.L.); 2Department of Biochemistry and Biotechnology, National Research Lobachevsky State University of Nizhny Novgorod, 23 Gagarin Avenue, 603022 Nizhny Novgorod, Russia; darazanegina@gmail.com (D.Z.); annbril@mail.ru (A.B.)

**Keywords:** salinity, *solanum*, root-to-shoot signaling, cytosolic pH, calcium signals, hydraulic signals, photosynthesis, transpiration, sodium, osmotic stress

## Abstract

Salinity is one of the most dangerous types of stress in agriculture. Acting on the root, salinity causes changes in physiological processes in the shoot, especially photosynthesis, which is crucial for plant productivity. In our study, we used potato plants, the most important crop, to investigate the role of salt-induced signals in changes in photosynthesis activity. We found a salt-induced polyphasic decrease in photosynthesis activity, and the earliest phase started several minutes after salt addition. We found that salt addition triggered rapid hydraulic and calcium waves from root to shoot, which occurred earlier than the first phase of the photosynthesis response. The inhibition of calcium signals by lanthanum decreased with the formation of rapid changes in photosynthesis. In addition to this, a comparison of the characteristic times of signal propagation and the formation of a response revealed the role of calcium waves in the modulation of rapid changes in photosynthesis. Calcium waves are activated by the ionic component of salinity. The salt-induced decrease in transpiration corresponds in time to the second phase of the photosynthetic response, and it can be the cause of this change. The accumulation of sodium in the leaves occurs a few hours after salt addition, and it can be the cause of the long-term suppression of photosynthesis. Thus, salinity modulates photosynthetic activity in plants in different ways: both through the activation of rapid distant signals and by reducing the water input and sodium accumulation.

## 1. Introduction

Salinity is one of the main problems in agriculture. Salinity is caused by both improper cultivation and irrigation [1] and climate change [2]. Salinity causes water stress, ionic imbalances, and osmotic and oxidative stress, which lead to decreased photosynthesis activity, slower growth, and yield decreases [3,4]. Salinity negatively affects yields because most crops are glycophytes [5]. Potato is the fourth most important crop in the world after rice, wheat, and maize, and it is also a glycophyte. Potato is an important part of the human diet, with it being a concentrated source of carbohydrates, proteins, and fats, as well as a variety of mineral elements and vitamins [6]. Salinity has more negative effects on potato than other crop plants, and it directly affects the quality and yield of tubers [7]. The yield of potato can be decreased by up to 60% due to growth in salty soil [8]. These facts prove the importance of research on the fundamental mechanisms of salinity’s effects that hinder the development of potential strategies for a decrease as a consequence of salt stress.

Salinity is a two-component stressor: on the one hand, it causes osmotic stress due to a decrease in water potential, which occurs due to sodium and chlorine ions moving outside the cell, and, on the other hand, it causes ionic toxicity, mainly due to Na^+^ [9]. The osmotic component of salinity mainly affects the water balance of the plant, which is why the transpiration limitations of photosynthesis are manifested [10]: stomata are closed due to a decrease in water input to the shoot and an increase in the synthesis of abscisic acid (ABA); as a result, the input of CO_2_ decreases [11,12,13]. The ionic component firstly disrupts, through sodium, the activity of a number of enzymes, including the Calvin–Benson cycle, chlorophyll biosynthesis, and enzymes that are included in the photosynthetic apparatus and thylakoid membranes [12,14,15]. Secondly, salinity leads to a change in the ratio of potassium/sodium ions, which affects the course of many metabolic potassium-dependent processes in the cell: the Calvin–Benson cycle, the phenylpropanoid pathway, glycolysis, and the synthesis of starch [9,16,17,18,19]. In studies on the effect of salinity on photosynthesis, a wide range of concentrations are used—from 20 mM to 400 mM NaCl—and all of them negatively affect photosynthesis [20,21,22,23,24,25,26,27,28,29].

Salinity causes a systemic response both in the root, which is in direct contact with salt, and in the shoot, due to the fact that it induces the propagation of signals [9,30]. Osmotic and ionic components are differently sensed and induce a signal, with them using different signaling systems [31,32]. The osmotic component is sensed by membrane tension receptor histidine kinase 1 (HK1) and mechanosensitive non-selective ion channels (NSCCs) [33]. The ionic component, even before loading sodium and chlorine into the cell, is sensed by glycosyl inositol phosphorylceramide (GIPC) and sphingolipids [34], as well as Feronia kinase [35]. Salinity can affect hydraulic signals, calcium, and reactive oxygen species (ROS) waves, as seen in rice and *Arabidopsis* [36,37,38,39], as well as changes in cytosolic pH, as also seen in rice and *Arabidopsis* [40,41,42,43], and electrical potentials, as seen in sunflower [44], as additional links in signal transmission.

The early changes in photosynthesis and the pathways that modulate it are currently poorly described for crop plants. Previously, we described fast salinity-induced effects in tobacco and potato plants [22]. However, the modulation of salt-induced changes in the activity of physiological processes and the possible role of root-to-shoot signals remain unexplored. The purpose of this study was to determine the possible role of salt-induced distant signals, including calcium, electrical, and hydraulic waves, in the induction of early changes in photosynthetic activity in potato plants.

## 2. Results

### 2.1. Salinity Caused a Decrease in Photosynthesis Activity and Transpiration

Salinity causes the inhibition of photosynthesis in potato leaves. We investigated changes in non-photochemical quenching (NPQ), photosystem II (PSII) fluorescence in the dark-adapted state (F_v_/F_m_)_,_ and the quantum yield of PSII Φ_PSII_ in three leaves of different strata. Salinity caused by 100 mM and 200 mM NaCl led to an increase in NPQ and a decrease in Φ_PSII_ in the leaves of all three strata. Different concentrations of NaCl cause the inhibition of photosynthesis to different degrees. A higher concentration (200 mM) led to a significant decrease in F_v_/F_m_ and an earlier significant increase in NPQ compared to a lower concentration (Figure 1).

The response of Φ_PSII_ and NPQ in dynamics in the first 2 h after the start of salt treatment in the leaf of the middle of the studied strata (Figure 2) with a high time resolution was determined for a more detailed study of the effects of salinity. As a result, NPQ in response to salt treatment was significantly increased after 24 and 26 min of 200 mM and 100 mM NaCl treatment, respectively. A large difference in the effect of NaCl at different concentrations was also identified. A significant decrease in Φ_PSII_ was observed at 110 min of 200 mM NaCl treatment, while 100 mM NaCl had not yet had an effect. NPQ significantly increased 10 min earlier when treated with a higher concentration of salt.

Figure 1 and Figure 2 show that the change in photosynthesis activity occurs in two phases: the short phase begins at no later than 10 min and lasts 30–40 min, the long phase begins after 60–70 min and lasts until the end of measurement, and a third phase is also possible, which begins after 3–4 h. Such polyphase dynamics are characteristic of the NPQ, reflecting the activity of the reactions of the light-dependent stage of photosynthesis. F_v_/F_m_, which reflects the intactness of the photosynthetic apparatus, is characterized by a monotonous decrease that occurs only when exposed to a high concentration of salt. The decrease in F_v_/F_m_ occurred 6 h after the start of salt treatment. There were no significant differences in changes in photosynthesis activity between the leaves of different strata when treated with 100 mM salt (Appendix A), but they were present in leaves 1 and 2 in F_v_/F_m_ after 8 h of 200 mM NaCl treatment (Appendix A).

At the same time, a decrease in transpiration was observed. Transpiration was determined using the Crop Water Stress Index (CWSI). At later observation periods (2–8 h of salt treatment), a higher concentration of NaCl had a stronger effect (Figure 3), but no significant differences were observed between different leaves (Appendix A). At the same time, during the first 120 min of observation, both salt concentrations had a similar effect.

Salinity had a negative effect on photosynthesis and transpiration. The inhibition was more severe when higher concentrations were used. The first changes in photosynthesis activity began a short time after treatment.

### 2.2. Salinity-Induced Ionic and Osmotic Stress Affects Photosynthesis and Transpiration

The effect of salinity on photosynthesis activity may be due to the accumulation of sodium and chlorine ions in the leaves. We measured the concentrations of ions in the leaves to verify this fact. The content of sodium ions in the leaves significantly exceeded the control level only 6 h after the start of salt treatment (Figure 4A), and the same level of accumulation was observed when plants were treated with both concentrations of NaCl (Figure 4B). The concentration of chlorine ions did not change significantly (Figure 4A).

This result indicates that the accumulation of ions in the leaves could not affect the early changes in photosynthesis activity. Other possible mechanisms of the salt effect include osmotic and ionic stress, which is mediated by NaCl in the roots, and the signal of which can be transmitted from the root to the shoot. We separated these components of salt stress. To simulate only the osmotic component of stress, an osmolyte, namely 200 mM sorbitol, was used, and 100 mM KCl was also used to test which ion—sodium or chlorine—caused signal induction under ionic stress. No significant differences were found in the effect of NaCl, KCl and sorbitol on the parameters of photosynthesis activity (Figure 5) and also in different leaves (Appendix A). The exception was the greater effect of NaCl 8 h after treatment, manifested as a pronounced increase in NPQ.

The effect of high concentrations of agents (200 mM NaCl, 400 mM sorbitol and 200 mM KCl) is shown in Appendix A.

Measurements of the photosynthesis activity with a high time resolution over a two-hour period showed that sorbitol and KCl, like NaCl, caused the rapid suppression of photosynthesis. At the same time, in the case of sorbitol treatment, significant differences compared to the control occurred after 21 min (Figure 6).

High concentrations of NaCl and sorbitol led to a decrease in Φ_PSII_ and an increase in NPQ, but KCl caused only statistically insignificant changes in Φ_PSII_ and NPQ (Appendix A).

Along with studying the effect of various agents on photosynthesis, their effect on the transpiration rate was assessed. Like NaCl, sorbitol and KCl caused a decrease in the intensity of transpiration (Figure 7). A comparison of the effects of various agents on transpiration during 120 min of treatment revealed a stronger and earlier effect of sorbitol compared to other agents. There was no difference in the earlier start of effects in the doubled concentrations of agents (Appendix A). No significant differences were found in the leaves of different strata when treated with singular and doubled concentrations of agents (Appendix A).

Eventually, the use of NaCl, sorbitol and KCl revealed their unidirectional effect on photosynthesis activity and transpiration. Among the main differences, some features can be noted. NaCl caused the greater inhibition of photosynthesis at a long time (8 h) after treatment. The effect of sorbitol on photosynthesis activity and transpiration began somewhat earlier than the effect of NaCl and KCl. The effect of KCl was weaker compared to other agents.

### 2.3. Salt-Induced Signals and Their Possible Role in Modulation of Photosynthesis Activity

The presence of a fast phase in the photosynthesis response indicates the propagation of a signal from the root to the shoot. The hydraulic signal induced by a change in the osmotic properties of the solution around the roots can directly modulate the stomata status. To observe the propagation of the hydraulic wave, we measured the change in the width of the stem, which may indicate a change in the flow of water through it. The width of the stem near leaf 2, on which the dynamics of NPQ and Φ_PSII_ were measured with a high time resolution, changed 3–5 min after the addition of NaCl, KCl or sorbitol (Figure 8). The same changes in the stem width were observed when treated with high concentrations of agents (Appendix A).

Another signal of a physical nature is a change in the electrical potential. We detected changes in the electrical potential in the root caused by various agents and monitored the propagation of the signal along the stem (Figure 9).

NaCl treatment caused hyperpolarization in the root. However, no propagation of the electrical signal in the shoot was observed. Sorbitol and KCl caused weak depolarization, which was more pronounced upon KCl treatment. Sorbitol, like NaCl, did not induce the propagation of an electrical signal in the stem. KCl treatment caused the propagation of electrical impulses in the stem (Appendix A). Similar changes were caused by treatment with high concentrations of agents (Appendix A).

Another possible distance signal affecting photosynthesis activity is calcium waves from the root to the shoot. NaCl caused a propagating calcium wave (Figure 10A). Osmotic stress caused by sorbitol also induced a weak calcium wave (Figure 10B).

In addition to the visualizing Ca^2+^ waves, their possible role in inhibiting photosynthesis activity was studied, and the calcium wave was locally blocked by lanthanum (La^3+^) (Figure 11). La^3+^ had a greater effect on the severity of phase 1 and the rate of the start of changes in photosynthesis in phase 2 upon NaCl treatment than upon sorbitol or KCl treatment (Figure 11). La^3+^ pretreatment caused significant differences compared to non-pretreated plants after 26 min of salt treatment. La^3+^ had a similar effect on reactions caused by high concentrations of agents within 2 h after the start of treatment (Appendix A).

Over a long observation period (2–8 h), no significant differences in photosynthesis responses were detected upon La^3+^ pretreatment (Figure 12, Appendix A). The inhibition of calcium signaling also had no significant effect on transpiration (Figure 11, Appendix A).

Finally, blocking the calcium signal had the greatest effect on the formation of the first rapid phase of the photosynthetic response, and especially upon NaCl treatment.

### 2.4. The Role of Cytosolic pH in Salinity

Changes in cytosolic pH are considered as a possible inducer of changes in photosynthesis activity during the transmission of a distance signal. To clarify the role of changes in cytosolic pH in changes in photosynthesis activity in the leaves and roots, cytosolic pH changes in response to treatment with NaCl, KCl and sorbitol were studied. The pH in the leaf cells did not change during the entire observation period (Figure 13A). Salt-induced alkalization and then subsequent acidification (12 min after NaCl treatment) was observed in the root; significant differences were observed in the range of 8–14 min after NaCl addition. Neither KCl nor sorbitol caused cytosolic pH changes in the roots (Figure 13B).

High concentrations of agents led to a significant decrease in cytosolic pH in leaves 1 and 2 at 4 h after treatment, and NaCl had a more powerful effect than sorbitol (Appendix A). In addition, there were significant differences in leaves of different strata after 8 h upon 200 mM NaCl and 400 mM sorbitol treatment, but not at low concentrations of agents (Appendix A). However, neither the reduced nor the increased concentrations affected the pH value in the leaves during the 120 min of treatment. The cytosolic pH also did not change in the stem zones where the propagation of the calcium wave was observed (Appendix A). The inhibition of calcium signals also had no effect on the cytosolic pH in the leaves and stem (Appendix A); however, the inhibition of alkalization in the roots was observed, and significant differences were observed compared to non-pretreated plants in the range of 8–14 min after NaCl addition (Figure 13C).

## 3. Discussion

### 3.1. Effects of Salinity on Physiological Processes in the Leaf

The results of this work demonstrate the suppressive effect of salinity on the activity of photosynthesis and transpiration in the leaves of potato plants. This fact is consistent with the previously described negative effects of salt of a similar concentration range on other plants [20,42,45,46]. Both salt concentrations (100 and 200 mM) used in the experiments caused unidirectional changes, but the amplitude of the changes was greater when the higher concentration was used. In particular, 100 mM NaCl during the experiment caused changes only in Φ_PSII_ and NPQ, which correspond to the efficiency of light energy use and the functioning of the electron transport chain, and 200 mM NaCl additionally caused a decrease in F_v_/F_m_, characterizing the integrity of the photosynthetic apparatus.

Since the process of photosynthesis begins long before Na^+^ accumulates in leaves, another explanation for the observed salt-induced effect is needed. One of the most obvious ways that salinity affects photosynthesis is through a decrease in transpiration caused by changes in water status [10], which occurs due to stomatal closure [47]. Stomatal closure may be mediated by ABA, the synthesis of which increases when water influx is reduced [48,49,50]. The decrease in CO_2_ entry caused by stomatal closure negatively affects the functionality of the Calvin–Benson cycle and, as a consequence, the light-dependent reactions of photosynthesis. Significant changes in CWSI, corresponding to the intensity of transpiration, occurred later than changes in photosynthesis activity and were likely responsible for the second phase of the photosynthesis response (Figure 14). Consequently, decreased transpiration may be an inducer of the second phase of the photosynthetic response, but it is unlikely that decreased transpiration is the cause of the earliest changes in photosynthesis.

The results of this study, which describe changes in photosynthesis activity preceding a decrease in transpiration rate and Na^+^ accumulation in the leaves, indicate the presence of a salt-induced root-to-shoot signal. This signal can be induced in root cells either directly by Na^+^ or by reducing the osmotic water potential of the solution bathing the roots, or by both [9]. To study the effect of each component of salinity, various agents were used: NaCl (osmotic and ionic components), sorbitol of equivalent osmolarity (osmotic component only) and KCl (osmotic and ionic components, absence of Na^+^ and presence of Cl^−^) [24,25,26]. Further analysis was based on a comparison of the parameters of distance signals and responses in the leaf upon treatment with each of the stimuli.

### 3.2. Salt-Induced Root-to-Shoot Signals

Ca^2+^, electrical and hydraulic signals were studied under the action of various agents. Calcium waves were studied using only osmotic (sorbitol) and complex (NaCl) stressors. In response to NaCl (ionic and osmotic components), a more pronounced and well-propagating calcium wave was detected in the stem (Figure 10). A calcium wave in the stem occurred almost immediately after the treatment of the roots and preceded the development of the first phase of the photosynthesis response. The formation of this phase can be associated with a calcium wave.

The propagation of a fast salt-induced Ca^2+^ wave in the shoot and an osmolyte-induced Ca^2+^ wave with a lower amplitude was demonstrated in other studies [51,52]. The root-to-shoot Ca^2+^ signal can be induced in various ways: hyperosmolarity can activate mechanosensitive NSCCs and osmosensor reduced-hyperosmolality-induced-[Ca^2+^]_i_-increase 1 (OSCA1) [53,54], as well as NSCCs through the generation of ROS by respiratory burst oxidase homologue (RBOH), which is activated by receptor kinase Feronia via Rho-of-plants (ROP) [55,56]. Na^+^ can activate NSCCs by binding to GIPC sphingolipids [34,57] or activating Feronia or Theseus1/Hercules [35]. Annexins can also be activated by ROS generated by RBOH, which can be activated by both components of salinity [58,59,60,61].

In addition to calcium waves, changes in the width of the stem, indicating changes in water mass flow, were studied using all agents. A decrease in the width of the stem was observed within a few minutes after the addition of agents to the root and was a later event than changes in water flow caused by changes in the water potential of the solution bathing the roots. Changes in water potential induce a hydraulic signal—a wave of changes in hydrostatic pressure [9]. The hydraulic signal is the fastest long-distance signal in plants, the speed of which exceeds the speeds of other types of signals [62,63].

The analysis of a long-distance signal of another type—electrical—showed changes in electrical activity directly in the area of action of the agents. NaCl caused hyperpolarization in the root; KCl and sorbitol caused depolarization. Previously, other researchers reported that salinity caused depolarization in tobacco [64] or hyperpolarization in *Arabidopsis* [65], and sorbitol caused hyperpolarization in tobacco [64] and *Arabidopsis* [66]. Our results may indicate species-specific features of the reaction in different plants [67]. The presence of changes in the electrical activity of root cells was not strictly related to the propagation of the electrical signal in the shoot. Pronounced electrical impulses in the stem, periodically occurring after the addition of agents, were detected only when treated with KCl. Our results do not confirm the decisive role of electrical signals as inducers of changes in photosynthesis activity during salinity.

Simultaneously with the hyperpolarization in the root cells, the cytosol was alkalized and then acidified (Figure 13A). This fact may indicate a short-term increase in the activity of plasmalemma H^+^-ATPase [68,69], which induced hyperpolarization in the root and calcium influx through voltage-gated ion channels [65,70,71]. H^+^-ATPase can be activated both as part of the activation of the salt overly sensitive (SOS) signaling pathway of Na^+^ exclusion from the cell [61] and by reducing the phosphoinositol concentration due to the activation of phosphatidylinositol-4-kinase (PI4K) [72,73]; both pathways are likely to involve Ca^2+^. The role of Ca^2+^ is confirmed by the suppression of the alkalization of the cytosol of root cells due to the inhibition of calcium influx by lanthanum. Further acidification of the cytosol of root cells is most likely due to the activation of Feronia when it senses Na^+^ in the cell wall [74] or is provoked by hyperosmolarity [55,56]. It is likely that cytosol acidification is stable during long-term salinity [46,75]. However, changes in cytosolic pH in the shoot occurred later and were caused by high salt concentrations. Therefore, changes in the cytosolic pH in the root can be considered only as one of the mechanisms serving to enhance calcium signaling in the root; in the shoot, it can be a consequence of Na^+^ accumulation.

Thus, only the hydraulic signal caused by the osmotic component of salt stress and the calcium wave, the propagation of which, according to the literature, is associated with the transmission of the ROS wave [76], are the most likely inducers of early changes in photosynthesis activity.

### 3.3. The Role of Long-Distance Signals in the Induction of Changes in Photosynthesis Activity

The analysis of the characteristic parameters of long-distance signals induced by various agents allows us to analyze the role of a certain signal in the induction of different phases of the photosynthesis response. First of all, it is necessary to note the stronger inhibition of photosynthesis upon NaCl treatment compared to other agents. However, a stronger photosynthetic response, expressed as a dramatic increase in NPQ (Figure 5), occurred 8 h after treatment and can be explained by the Na^+^ accumulation in the leaf and the direct effect of Na^+^ on photosynthetic cells. Early changes in photosynthesis activity parameters were comparable only for osmotic (sorbitol) and complex (NaCl) agents (Figure 5 and Figure 6). Our results are consistent with previous studies showing similar salt- and osmolyte-induced effects on photosynthesis [24,25,77]. This fact may indicate an important role of the osmotic component of salt stress in the generation of long-distance signals, causing a decrease in photosynthesis activity.

At the same time, osmotic or complex stress-induced differences in the mechanisms of induction of the photosynthetic response should be noted. In particular, calcium wave suppression by lanthanum highlighted these differences. Firstly, lanthanum did not have a significant effect on long-term changes in photosynthesis. There was also no effect on Na^+^ accumulation (Appendix A). Lanthanum had an effect on the first fast phase of the photosynthesis response. This fact confirms the role of the Ca^2+^ wave in the induction of this phase. The effects of different agents—NaCl and sorbitol—depended differently on La^3+^ pretreatment. Lanthanum had a stronger effect on the salt-induced response. In combination with the salt-induced long-distance calcium wave, this fact confirms the significant contribution of the calcium wave induced by the ionic component of salt stress in the roots to the induction of the first fast phase of the photosynthesis response.

Ca^2+^ can inhibit photosynthesis by binding to the chloroplast sensory protein CAS, which is crucial for energy-dependent quenching, an energy-dependent component of NPQ that regulates the thermal dissipation of excess absorbed light energy [78]. Previous works suggest [79,80] that Ca^2+^ influx into chloroplasts most likely occurs through the mechanosensitive channels of small conductance-like proteins (MSL) and probably Ca^2+^/heavy metal pumps HMA1 and Ca^2+^-ATPases ACA1. Later changes in photosynthetic activity can be explained by the effect of Ca^2+^ on fructose-1,6-bisphosphatase [81] and, in general, on the functionality of the Calvin–Benson cycle [82].

The absence of the effect of calcium wave suppression on long-term changes in photosynthesis does not mean that Ca^2+^ is not involved in the regulation process directly in the leaf. An increase in calcium levels in photosynthetic cells can be induced by a hydraulic signal, which can be sensed in the leaves by mechanosensitive channels [83], as well as other possible receptors, but their role has not been proven [84]. Hydraulic waves can also cause an increase in ROS levels [85]. ROS trigger the production of ABA, which stimulates stomatal closure [84].

Additionally, the decrease in water mass flow in leaf cells causes the opening of PIP aquaporins, the increased synthesis of ABA and the activation of anion channels SLAC1 and K^+^ channels GORK and stomatal closure [48,49,50].

The reduction in CO_2_ availability that occurs due to stomatal closure leads to a decrease in ribulose-1,5-biphosphatase carboxylase/oxygenase activity, which also affects light-dependent photosynthetic reactions in the later phases of photosynthesis changes [48].

Another possible mechanism involved in modulating photosynthesis activity, Ca^2+^-related or Ca^2+^-unrelated, is a change in cytosolic pH. Previously, the possible role of the cytosolic pH, closely linked to the Ca^2+^ signaling system, in salt-induced signal transmission was reported [40,41]. However, the leaf cytosolic pH began to change later than photosynthesis and transpiration, only upon treatment with high concentrations of agents (Appendix A), whereas there were no significant changes upon treatment with low concentrations of agents (Figure 13). The cytosolic pH in the leaf was also unaffected by calcium signal inhibition in the stem (Appendix A). It can be assumed that the observed changes in cytosolic pH are associated with the accumulation of Na^+^ in the leaves. Therefore, the proton signaling system is not an intermediate link between Ca^2+^ signals and photosynthesis, and the pH does not affect the first phases of changes in photosynthesis activity.

Thus, early changes in photosynthesis activity are associated with both the ionic component of salinity and the osmotic component. The osmotic component mainly triggers a hydraulic wave, and the ionic component triggers calcium signals. Probably, in the leaf, signals from both components trigger the calcium regulation of photosynthesis and modulate the first phase of photosynthesis. Subsequent phases of changes in photosynthesis activity are associated with a decrease in water flow and the closure of stomata, and then with the accumulation of Na^+^ in the leaves.

## 4. Materials and Methods

### 4.1. Plant Material and Treatment

Plants of *Solanum tuberosum* (variety Nevsky) were used in the experiments. Potato plants constitutively expressing the fluorescent pH-sensitive sensor Pt-GFP [86] were also used to measure the cytosolic pH. The procedure for the agrobacterial transformation of potato was described in a previous study [87]. Potato microplants were obtained by in vitro propagation, in vitro cultivation after 4 weeks and adaptation to ex vitro conditions after 1 week. Ex vitro adaptation was carried out in hydroponic pots with tap water; for the first 3 days, the plants were covered with glass pots. The microplants were cultivated in a growth room at 24 °C with a 16-h light/8-h dark cycle and cool-white luminescent tubular lamps (OSRAM, Munich, Germany; 60 µmol/m^2^ s).

To inhibit the increase in cytosolic Ca^2+^ levels, the plasma membrane Ca^2+^-permeable channel blocker lanthanum (LaCl_3_*7H_2_O) was used. A part of the stem (2.3 cm) was placed in a plastic vessel with 1 mM La^3+^ solution at 2 h before the salt/osmotic treatment and during the salt/osmotic treatment.

Salt stress was caused by treatment with 100 mM or 200 mM NaCl and 100 mM or 200 mM KCl; osmotic stress was caused by treatment with 200 mM or 400 mM sorbitol. The treatment with a solution without NaCl, KCl or sorbitol (tap water) was used as a control.

### 4.2. Chlorophyll Fluorescence Measurements

The parameters of light-driven reactions of photosynthesis were recorded using PlantExplorerPro+ (PhenoVation, Wageningen, The Netherlands) and IMAGING-PAM MINI Version (Heinz Walz GmbH, Effeltrich, Germany). The parameters of photosystem II fluorescence in the dark-adapted state (F_v_/F_m_), the quantum yield of photosystem II photochemical reactions (Φ_PSII_) and non-photochemical fluorescence quenching (NPQ) were calculated using Data Analysis Software Version 5.6.7-64b (PhenoVation, Wageningen, The Netherlands) or ImagingWin v2.41a FW MULTI RGB (Heinz Walz GmbH, Effeltrich, Germany) of the device according to the following equations [88]:F_v_/F_m_ = (F_m_ − F_0_)/F_m_,(1)
Φ_PSII_ = (F_m_′ − F)/F_m_,(2)
NPQ = (F_m_ − F_m_′)/F_m_′,(3)
where F_0_ is the image of the minimum fluorescence of chlorophyll in the dark-adapted state, F_m_ is the maximum fluorescence yield, F_m_′ is the maximum fluorescence yield in the light condition and F is the current fluorescence level.

White actinic light was used to maintain photosynthesis, and the light was the sum of photon flux from diodes with emission maxima at 455, 525 and 660 nm. The photon flux density of the actinic light was 136 (PlantExplorerPro+) or 270 (IMAGING-PAM MINI Version) µmol/m^2^ s. Saturation pulses were generated by illumination at a wavelength of 635 nm with a photon flux density of 2881 µmol/m^2^ s. The duration of saturation pulses was 300 ms. The duration of the dark adaptation preceding measurements was 15 (PlantExplorerPro+) or 30 (IMAGING-PAM MINI Version) minutes. The acquired images were processed using Data Analysis Software Version 5.6.7-64b or ImagingWin v2.41a FW MULTI RGB.

### 4.3. Fluorescent Imaging of Cytosolic pH and Ca^2+^

Fluorescent imaging of the cytosolic pH in potato leaves was carried out by recording changes in the fluorescence of the Pt-GFP sensor. The fluorescence of Pt-GFP was excited at 445–475 nm and 390–420 nm in PlantExplorerPro+ (PhenoVation, Wageningen, The Netherlands) using luminodiodes; the emitted fluorescence was captured using a CMOS sensor with a 530 ± 20 nm filter. Fluorescent images were acquired at an exposure level of 250 ms. The dependence of Pt-GFP fluorescence on the pH in the leaves was analyzed to determine the cytosolic pH levels. To determine this dependence, leaves were incubated in buffer solutions with different pH levels from 4 to 9 (0.5 increments) and 125 µM of the ionophore carbonyl cyanide 3-chlorophenylhydrazone (CCCP). The composition of the buffer solutions is described in [87]. The incubation time was 4 h. After incubation, images of leaf fluorescence were acquired; the images were processed using Data Analysis Software Version 5.6.7-64b and FiJi [89] to calculate the dependence of Pt-GFP fluorescence on the cytosolic pH.

The cytosolic pH in the roots was also visualized using a Pt-GFP sensor on the fluorescence imaging system DVS-03 (ILIT RAS, Shatura, Russia). The fluorescence of Pt-GFP was excited by a 395/25 nm luminodiode and 490/20 nm luminodiode and emitted by a CMOS camera (PRIME 95B, Photometrics, Tucson, AZ, USA) with a 535/43 nm filter. Fluorescent images were captured at an exposure level of 2000 ms every 30 s.

Cytosolic Ca^2+^ levels were visualized using the Fluo-4, AM probe (Molecular Probes, Carlsbad, CA, USA) in non-transgenic plants and the fluorescence imaging system DVS-03 (ILIT RAS, Shatura, Russia). A part of the stem (2.3 cm) was placed in a plastic vessel with 20 μM Fluo-4, AM solution, and then the probe solution was loaded three times through the stem using a desiccator and a vacuum pump at pressure of 0.2 atmospheres for 5 min. Incubation in the probe solution lasted 1 h. Fluorescence was excited by a 490/20 nm luminodiode and emitted by a CMOS camera (PRIME 95B, Photometrics, Tucson, AZ, USA) with a 535/43 nm filter. Fluorescent images were captured at an exposure level of 2000 ms every 10 s.

The Micro-Manager 1.4 software [90] and ImageJ 1.52p software [91] were used to process the raw images obtained with DVS-03.

### 4.4. CWSI Measurements

The monitoring of changes in the CWSI of potato leaves was performed by determining the leaf surface temperature (T) and the temperatures of moisture (T_moisture_) and dry (T_dry_) standards using a thermal imager, the Testo 885 (Testo, Lenzkirch, Germany). The acquired images were processed using the IRSoft software (version 4.8) (Testo, Lenzkirch, Germany), and the stomatal conductance value of the leaf was obtained using the equation [92]
CWSI = (T_dry_ − T)/(T_dry_ − T_moisture_).(4)

### 4.5. Hydraulic Signal Measurements

Hydraulic signals were detected by changes in stem width. The stem width was measured by a system including laser distance sensors (SICK, Düsseldorf, Germany), an evaluation unit AOD1 (SICK, Düsseldorf, Germany) and analog-to-digital converters LTR12 in crate LTR-EU-2-5 (L-Card, Moscow, Russia). The plant was fixed on a plastic stand at the same distance between the sensor–transmitter and the sensor–receiver. The region of interest (Figure 8) was located horizontally and at the same level as the micrometer light path. The recording frequency was 10 Hz. Data were processed using the L Card Measurement Studio (LMS) software (version 1.1.0) (L-Card, Moscow, Russia).

### 4.6. Measurements of Electrical Potential Using Macroelectrodes

Electrical potential was measured extracellularly using Ag^+^/AgCl electrodes EVL-1MZ (Gomel Plant of Measuring Devices, Gomel, Belarus). The electrodes were connected to a high-impedance amplifier IPL-113 (Semico, Novosibirsk, Russia) and a personal computer. The measuring electrodes were connected with a plant using cotton threads wetted with water. The scheme of the placement of electrodes on the plant is shown in Figure 9. The reference electrode was placed on the top leaf in a Petri dish with tap water.

### 4.7. Measurements of Na^+^ and Cl^−^ Content

The Na^+^ content in leaves was determined using an Ag^+^/AgCl ion-selective glass electrode ELIS-112Na (NPO Measuring Technology MT, Moscow, Russia) and an Ag^+^/AgCl reference electrode ELIS-1M3.1 filled with a 3M KCl solution (NPO Measuring Technology MT, Moscow, Russia). The Cl^−^ content was determined using an ion-selective film electrode ELIS-131Cl (NPO Measuring Technology MT, Moscow, Russia) and Ag^+^/AgCl reference electrode EVL-1M3.1 filled with a 1M KNO_3_ solution (NPO Measuring Technology MT, Moscow, Russia). The electrodes were connected to the high-impedance amplifier IPL-112 (Semiko, Novosibirsk, Russia).

The leaves were dried for 48 h at 70 °C and ground into a powder. A standard weighed sample of dried leaves was dissolved in 25 mL of distilled water and heated at 90 °C for 3 h. The pH was then adjusted with ammonia vapors above 8 pH units and the concentrations of sodium and chloride ions were measured using electrodes. The ion content was expressed in mmol/g dry weight (DW).

### 4.8. Statistical Analysis

Statistical analysis was performed using the GraphPad Prism 6 software (GraphPad Software, Boston, MA, USA). Data are presented as the mean ± standard error of the mean (SEM). The number of biological replicates ranged from 5 to 9. The data were analyzed using the multiple *t*-test. * is *p* < 0.05.

## 5. Conclusions

The results show that salinity causes a decrease in photosynthesis activity, and the response of photosynthesis is characterized by multiple phases. The earliest changes, which begin 10–20 min after salt treatment, are modulated by rapid distance signals. Calcium and probably hydraulic waves play an important role in the induction of the earliest responses of photosynthesis. Calcium waves are triggered by the ionic component of salinity. The formation of the next, longer phase of the photosynthesis response is apparently associated with a decrease in the intensity of transpiration due to a decrease in water influx, as evidenced by the synchronicity of the observed changes. The latest phase of the photosynthesis response occurs after 3–4 h of salt treatment and is probably due to Na^+^ accumulation in the leaves.

The complexity of salt-induced changes reveals the great importance of integrating the response throughout the plant for stress adaptation. Root-to-shoot signals also play an important role in this process. Further studies should focus on the molecular mechanisms of salt-induced signals and the mechanisms of their propagation, as well as on decoding the methods of induction (by various signals) of functional responses in the shoot. These studies and their results will contribute to the development of optimal strategies to optimize the cultivation of important crops, including potato, under salinity conditions.

## Figures and Tables

**Figure 1 ijms-25-01229-f001:**
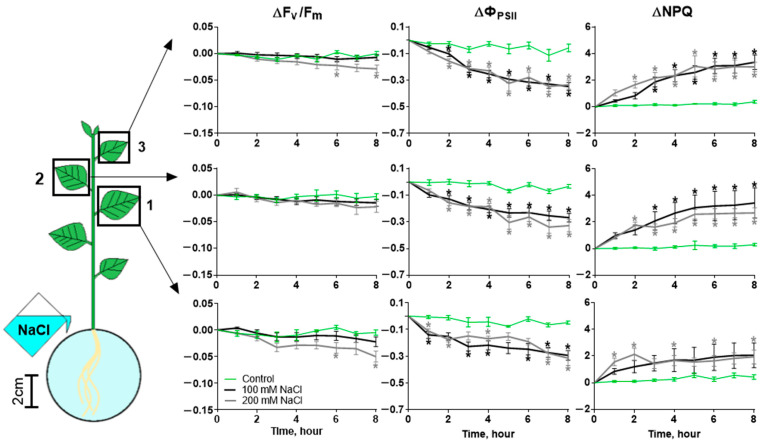
Changes in photosynthesis activity in three leaves (1, 2 and 3 in the scheme) of potato during NaCl treatment. Changes in F_v_/F_m_, Φ_PSII_ and NPQ were caused by 100 mM or 200 mM NaCl. Control comprised plants treated with water. Data present the difference in F_v_/F_m_, Φ_PSII_ or NPQ between time points before and after treatment. Data present the mean ± SEM (*n* = 9), asterisks (*) whose color corresponds to the line color indicate data significantly different (*p* < 0.05) from the control.

**Figure 2 ijms-25-01229-f002:**
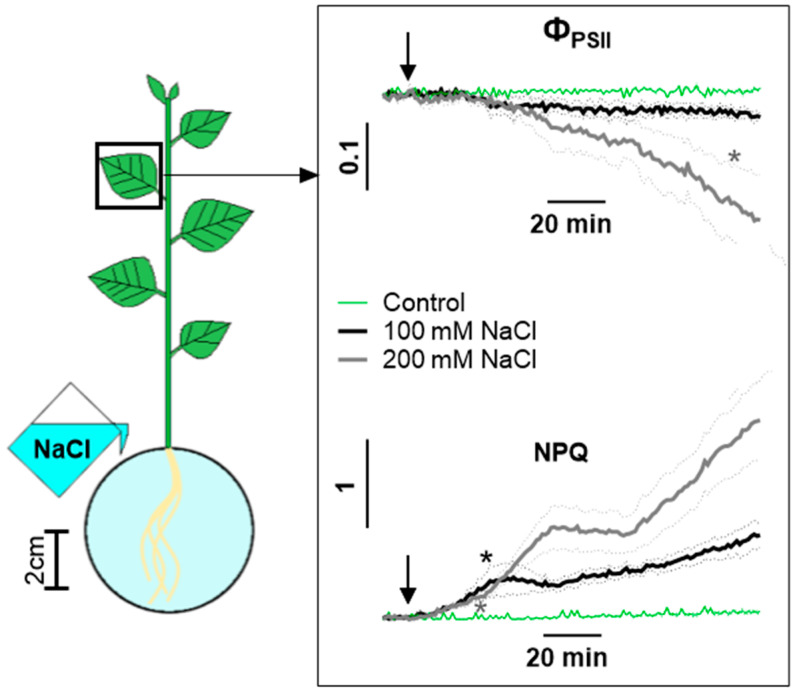
Changes in photosynthesis activity in leaf during 2 h of salt treatment. Changes in Φ_PSII_ and NPQ were caused by 100 mM or 200 mM NaCl. Control comprised plants treated with water. The arrow indicates the moment of treatment. Data present the mean ± SEM (*n* = 6), asterisks (*) whose color corresponds to the line color indicate first data significantly different (*p* < 0.05) from the control.

**Figure 3 ijms-25-01229-f003:**
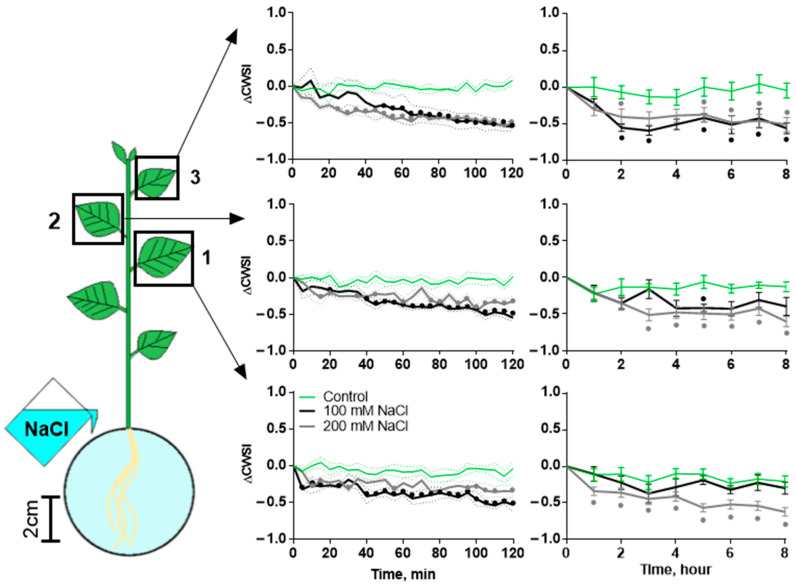
Changes in CWSI of three leaves (1, 2 and 3 in the scheme) during salt treatment. Changes in CWSI were caused by 100 mM or 200 mM NaCl. Control comprised plants treated with water. Data present the difference in CWSI between time points before and after treatment. Data present the mean ± SEM (*n* = 6), bullets (•) whose color corresponds to the line color indicate data significantly different (*p* < 0.05) from the control.

**Figure 4 ijms-25-01229-f004:**
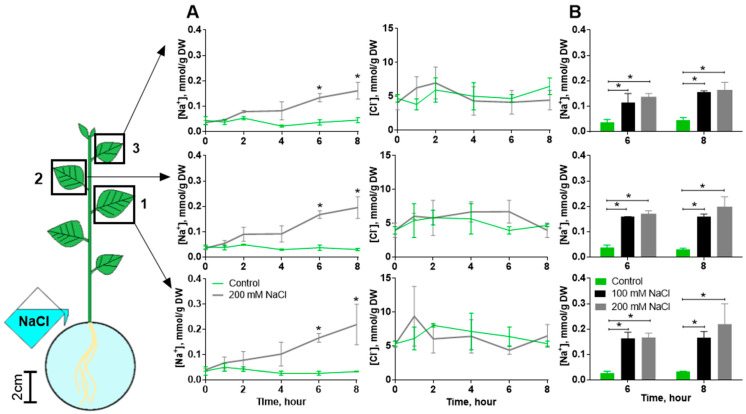
Accumulation of ions in three leaves (1, 2 and 3 in the scheme) of potato during salinity. (**A**) Dynamics of Na^+^ and Cl^−^ accumulation during 200 mM NaCl treatment. (**B**) Accumulation of sodium ions was similar after 6 and 8 h when treated with 100 mM and 200 mM NaCl. Control comprised plants treated with water. Data present the mean ± SEM (*n* = 4, where *n* is biological replication, which included 5–7 leaves), * *p* < 0.05 salt treatment versus control.

**Figure 5 ijms-25-01229-f005:**
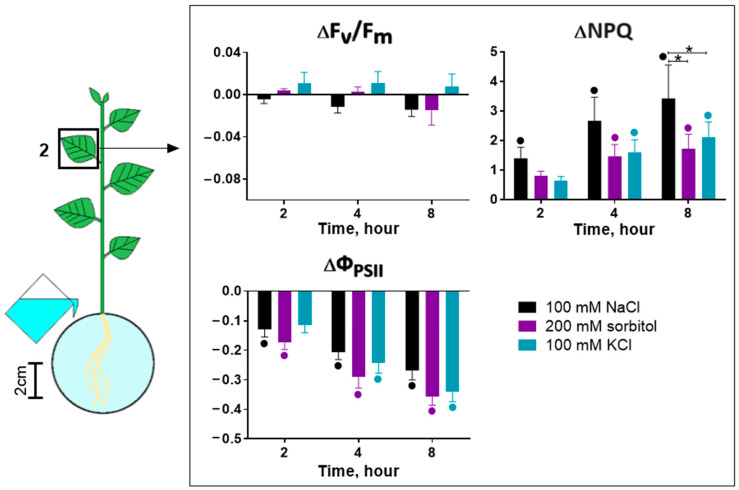
Effect of treatment with 100 mM NaCl, 200 mM sorbitol or 100 mM KCl on photosynthesis activity. ΔF_v_/F_m_, ΔΦ_PSII_ and ΔNPQ represent the difference in F_v_/F_m_, Φ_PSII_ and NPQ between treated and control plants. Data represent the mean ± SEM (*n* = 9), bullets (•) whose color corresponds to the column color indicate data significantly different (*p* < 0.05) from the control, * *p* < 0.05 NaCl treatment versus sorbitol or KCl treatment.

**Figure 6 ijms-25-01229-f006:**
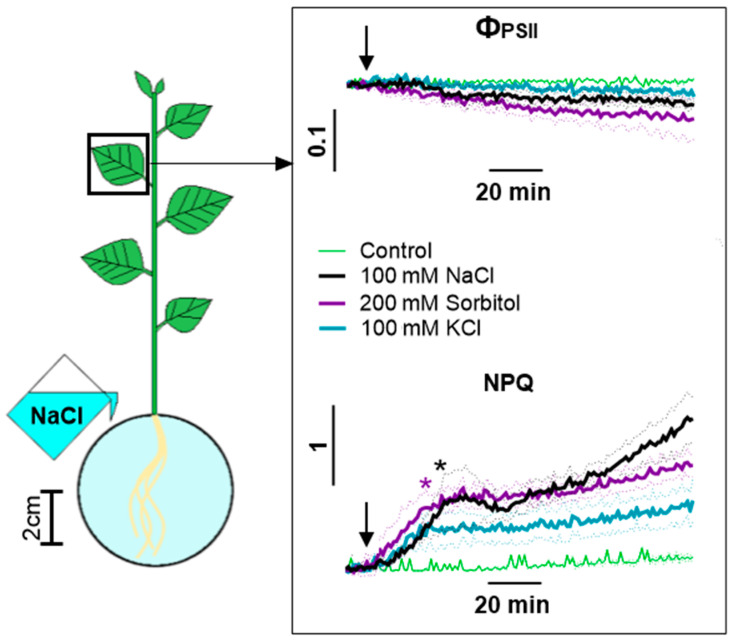
Changes in photosynthesis activity in leaf during 2 h of 100 mM NaCl, 200 mM sorbitol or 100 mM KCl treatment. Control comprised plants treated with water. The arrow indicates the moment of treatment. Data represent the mean ± SEM (*n* = 6), asterisks (*) whose color corresponds to the line color indicate first data significantly different (*p* < 0.05) from the control.

**Figure 7 ijms-25-01229-f007:**
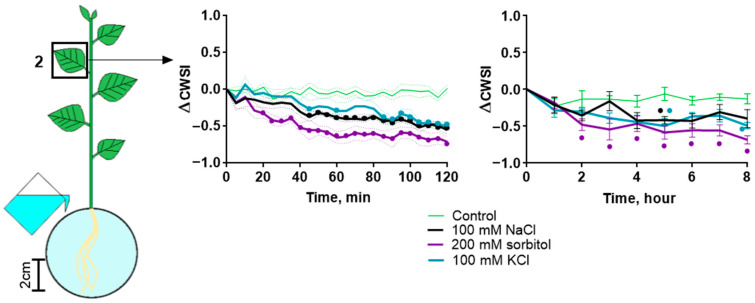
Changes in CWSI in leaf during 100 mM NaCl, 200 mM sorbitol or 100 mM KCl treatment. Control was plant treated with water. Data represent the difference in CWSI between time points before and after treatment. Data represent the mean ± SEM (*n* = 6), bullets (•) whose color corresponds to the line color indicate data significantly different (*p* < 0.05) from the control.

**Figure 8 ijms-25-01229-f008:**
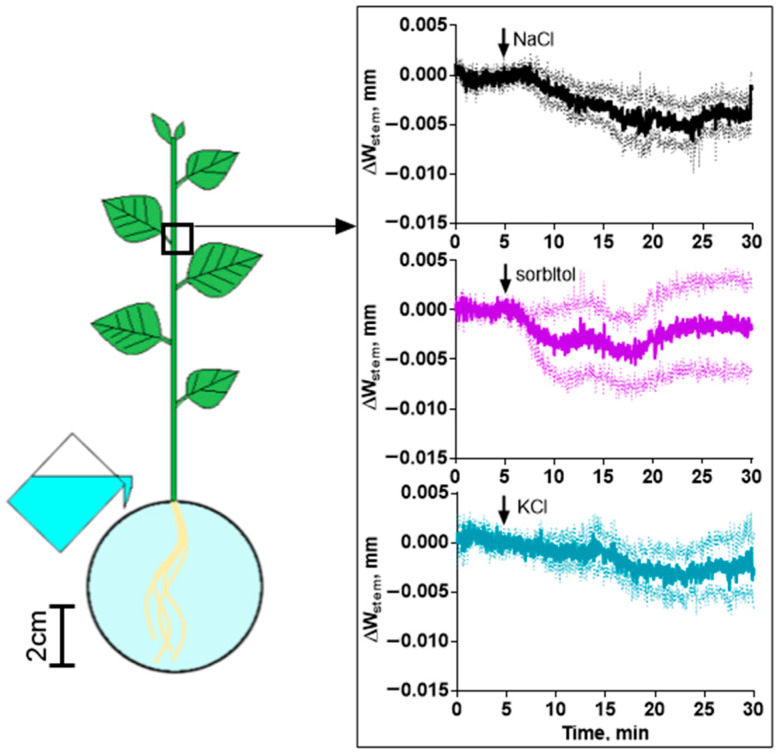
Changes in the width of the stem (ΔW_stem_) induced by 100 mM NaCl, 200 mM sorbitol or 100 mM KCl. The arrow indicates the moment of treatment. Data represent the mean ± SEM (*n* = 5).

**Figure 9 ijms-25-01229-f009:**
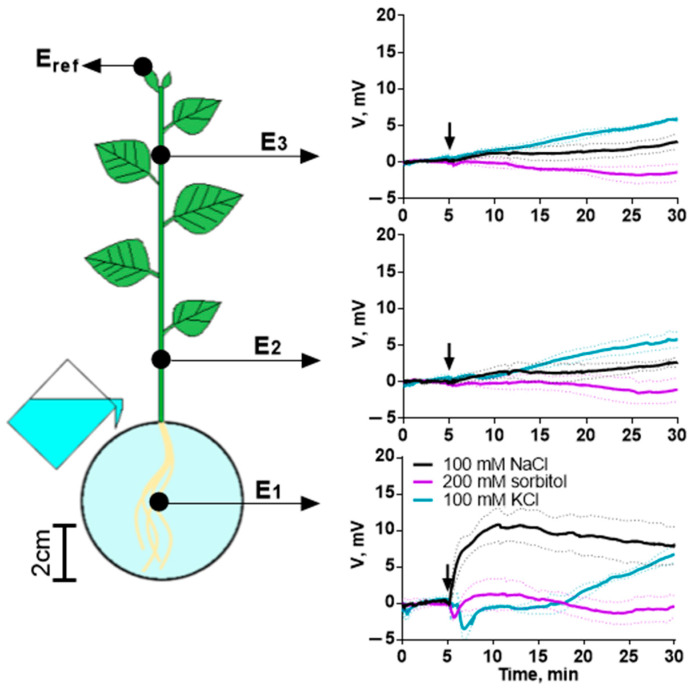
Changes in electrical potential (V) induced by 100 mM NaCl, 200 mM sorbitol or 100 mM KCl in potato stem and root. The placement of the measuring electrodes (E_1_, E_2_, E_3_) is shown by dots in the scheme. The reference electrode (E_ref_) is placed on the top leaf. The arrow indicates the moment of treatment. Data represent the mean ± SEM (*n* = 5).

**Figure 10 ijms-25-01229-f010:**
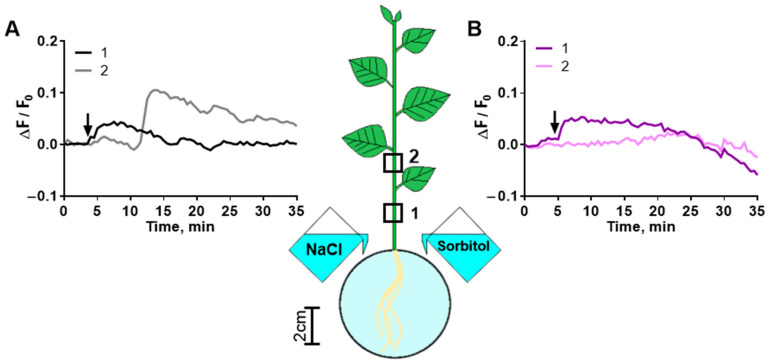
Typical dynamics of intracellular Ca^2+^ changes induced by NaCl (**A**) or sorbitol (**B**) in potato stem. Changes in Ca^2+^ concentration caused by 200 mM NaCl or 400 mM sorbitol were determined in two parts of the stem (1 and 2). Changes in Ca^2+^ levels were visualized using the fluorescent probe Fluo-4, AM. Changes in Ca^2+^ levels are presented as relative changes in Fluo-4 fluorescence (ΔF/F_0_). The arrow indicates the moment of treatment.

**Figure 11 ijms-25-01229-f011:**
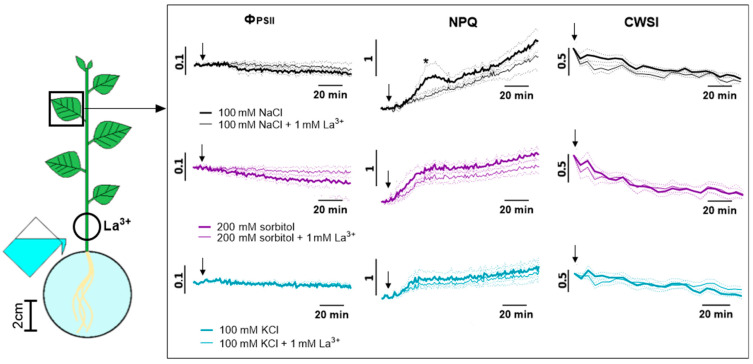
Effect of Ca^2+^ signal inhibition on photosynthesis activity and transpiration induced by 100 mM NaCl, 200 mM sorbitol or 100 mM KCl in the leaf. Ca^2+^ signal was inhibited by 1 mM La^3+^. The circle shows the part of the stem that was incubated in La^3+^ solution. The arrow indicates the moment of treatment. Data represent the mean ± SEM (*n* = 6), asterisks (*) whose color corresponds to the line color indicate first data significantly different (*p* < 0.05) of treatment without La^3+^ pretreatment from the control.

**Figure 12 ijms-25-01229-f012:**
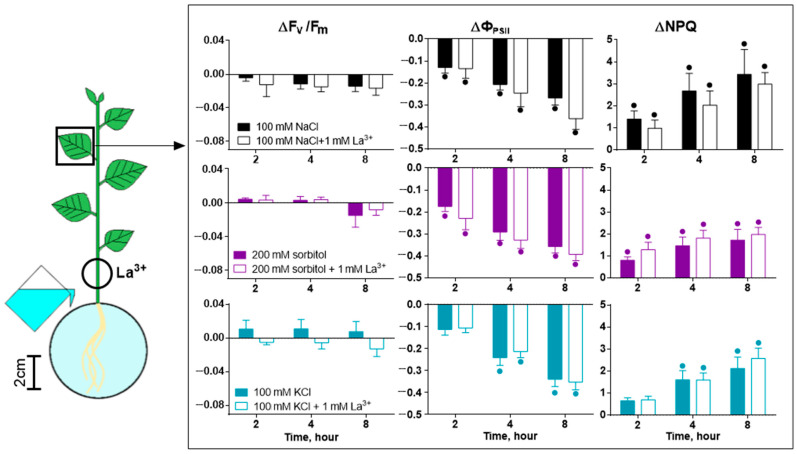
Effect of 100 mM NaCl, 200 mM sorbitol and 100 mM KCl on photosynthesis activity in the leaf with La^3+^ pretreatment. The circle shows the part of the stem that was incubated in La^3+^ solution. ΔF_v_/F_m_, ΔΦ_PSII_ and ΔNPQ represent the difference in F_v_/F_m_, Φ_PSII_ and NPQ between treated and control plants. Data represent the mean ± SEM (*n* = 9), bullets (•) whose color corresponds to the column color indicate data significantly different (*p* < 0.05) from the control.

**Figure 13 ijms-25-01229-f013:**
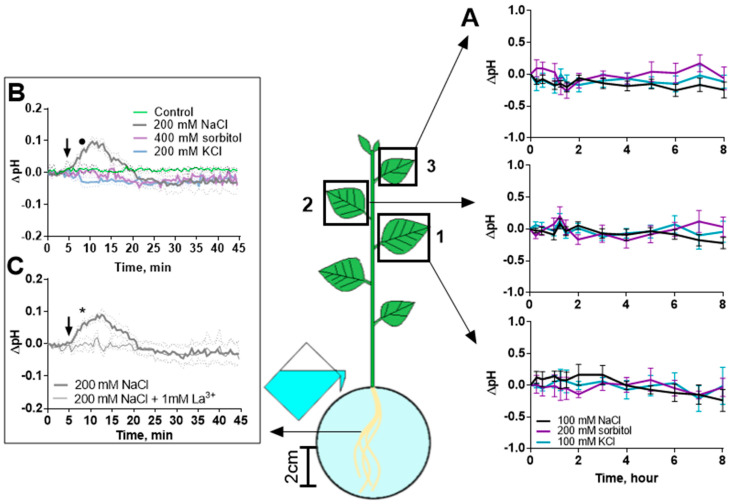
Effect of NaCl, sorbitol and KCl on cytosolic pH in leaves and roots of potato plants. Cytosolic pH in leaves and roots was measured using fluorescent probe Pt-GFP. (**A**) Dynamics of cytosolic pH in three leaves (1, 2 and 3 in the scheme) upon 100 mM NaCl, 200 mM sorbitol and 100 mM KCl treatment. (**B**) Dynamics of cytosolic pH in roots treated with 200 mM NaCl, 400 mM sorbitol or 200 mM KCl. Control comprised plants treated with water. (**C**) Pretreatment with La^3+^ leads to inhibition of NaCl-induced alkalization of cytosol in root cells. For (**A**), ΔpH represents the pH difference between treated and control plants. For (**B**,**C**), data represent the pH difference between time points before and after treatment, and the arrow indicates the moment of treatment. La^3+^ solution was added to the roots 2 h before NaCl treatment. Data represent the mean ± SEM (*n* = 9), bullet (•) indicates first data significantly different (*p* < 0.05) from the control, asterisk (*) indicates first data significantly different (*p* < 0.05) from the treatment with La^3+^ pretreatment.

**Figure 14 ijms-25-01229-f014:**
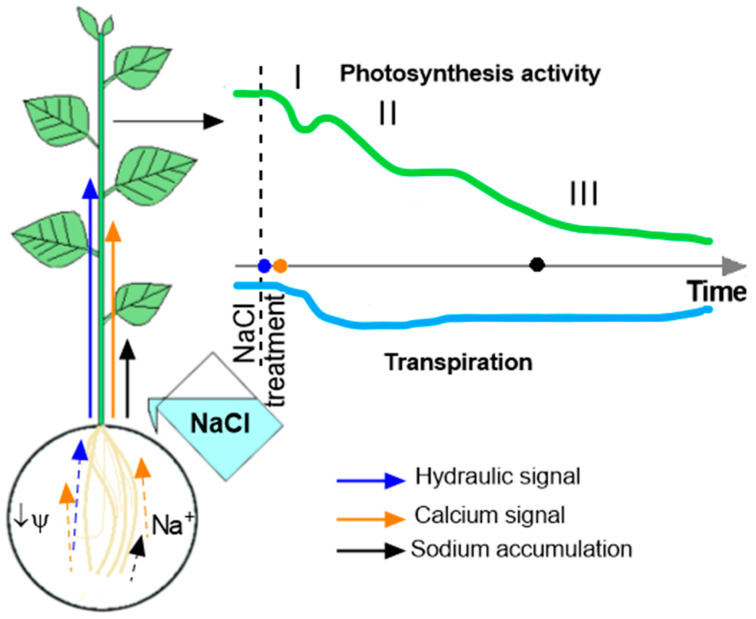
Dynamics of photosynthesis activity and transpiration during NaCl treatment. The generation of calcium and hydraulic signals occurs in roots directly exposed to salt stress. The ionic component (Na^+^) of salinity induces root-to-shoot calcium waves, and the osmotic component (Ψ) triggers root-to-shoot hydraulic signals. These signals modulate photosynthesis activity. Changes in photosynthesis activity are characterized by the presence of phases I, II and III. Root-to-shoot signals affect phase I; phase II is possibly modulated by a decrease in transpiration rate. Phase III of photosynthesis activity is probably associated with Na^+^ accumulation.

## Data Availability

Data are contained within the article and Appendix A.

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
