# Peer review of "Salt-Induced Early Changes in Photosynthesis Activity Caused by Root-to-Shoot Signaling in Potato"

_ijms, 2024, doi:10.3390/ijms25021229_

Round 1

Reviewer 1 Report

Comments and Suggestions for Authors

Review of the manuscript entitled "Salt-induced early changes in photosynthesis activity caused by root-to-shoot signaling". I find the work interesting and valuable. I have a few thoughts that require consideration before accepting the work for publication.

I have comments on the presentation of the results. It is not perfect. Sometimes control plants are included in the graphs and sometimes not. Sometimes the graphs show the dynamics of change and sometimes they show the results obtained at specific time points. It is best to adopt a uniform scheme for presenting results. It is worth asking whether the different stressors acted identically? Did the individual leaves tested (different leaf establishment height was part of the experiment?) differ statistically with respect to the parameters tested? I think that the presentation of results and statistical analysis should be worked on a bit. Perhaps you should use classical statistical software and not GraphPad Prism which you use to create graphs.

Specific comments:

  1. The title should include information on which plant the research was conducted.
  2. The abstract should be written in the classical structure: introduction, purpose and scope of the study, including the research methods used and the main results of the work. The current version is not perfect. It should be remembered that most readers of scientific papers read only the abstract.
  1. Keywords duplicate the information in the title of the article. This should be corrected.
  2. L426-441. On how many plants per research variant was the experiment conducted? On which plant organs was the research conducted? In how many replicates per research variant? The experiment should be described very precisely.
  3. L486. In the chapter Plant Material and Treatment, there is no information about the use of non-transgenic plants in the study. Please complete the description of the study material.
  4. CWSI Measurement: A method used to assess plant transpiration?
  5. L498. Was the study conducted on potatoes or tobacco?
  6. L510. The region of interest (Figure 8) was located horizontally and at the same level as the track of micrometer light.
  7. L519. The scheme of electrode positions on the leaf is shown in Figure 9.
  8. L538-541. What statistical test was used? I do not see the results of the two-way ANOVA analysis in the results presented. Please cite correctly the software used (Company, country, headquarters).
  9.  Fig. 4. Parts A and B of the graph are not compatible. A. shows control and 200nM NaCl and part B does not include control but shows 100 and 200 nM KCL. In part B - no statistical analysis.
  10. Fig. 5. The graph does not include control plants! There are no statistics.
  1. Fig. 6. No line for control plants? Statistics show differences from control?
  2. Fig. 7. No lines for control plants? Statistics show differences from control? The graph shows the dynamics of change. Were there no statistical differences between the test subjects at 6 and 8 o'clock compared to the control? Perform a factor analysis.
  3. Fig. 12. I am not sure which part shows La3+ pretreatment and which part shows a lack of La3+. What was the control? There are no control bars on the graph. No statistics.
  1. Fig. 13. The statistics are not clear on the graph. Were there statistical differences between the tested leaves (1, 2, and 3)?
  2. In Fig. 14, parts of the graph A, B, and C are not compatible. They do not show the same objects. No statistics. Were there statistical differences between 6 and 8 o'clock? Did the studied parameters for leaves 1–3 differ statistically?

Author Response

Response to Reviewer 1

Dear Reviewer,

We are grateful to you for your comments and suggestions for improving our work. We have revised the manuscript according to your comments. The presence of differences between stressors is shown in the graphs where this is the case. We have added information to the text of the manuscript about the absence/presence of differences for leaves of different strata. We also made additional figures in the Supplementary, and these figures help easy to compare the parameters in the leaves of different strata. Below are the answers to your questions mentioned in the review (the reviewer's comments are highlighted in bold, our answers follow in plain text).

Point 1: The title should include information on which plant the research was conducted.

Answer 1: We have added the object of research in the title of the manuscript.

Point 2: The abstract should be written in the classical structure: introduction, purpose and scope of the study, including the research methods used and the main results of the work. The current version is not perfect. It should be remembered that most readers of scientific papers read only the abstract.

Answer 2: Thank you for your comment! We have added to the Abstract information about the main results of the work. At the same time, we take into account the rules of the Journal, which limit abstract size by 200 words.

Point 3: Keywords duplicate the information in the title of the article. This should be corrected.

Answer 3: Thank you for your comment! We changed the list of keywords.

Point 4: L426-441. On how many plants per research variant was the experiment conducted? On which plant organs was the research conducted? In how many replicates per research variant? The experiment should be described very precisely

Answer 4: In the caption to each figure, there are information about the number of plants in each variation. Also, in the chapter Statistical Analysis in Methods, we indicated the range of replicates in all experiments.

Point 5: L486. In the chapter Plant Material and Treatment, there is no information about the use of non-transgenic plants in the study. Please complete the description of the study material.

Answer 5: Thank you for the comment! We used genetically transformed plants to measure pH. Other studies have used transgenic plants. In a preliminary series of experiments, it was shown that there were not differences in salt-induced photosynthesis responses for transgenic and non-transgenic potato plants. We have added relevant information.

Point 6: CWSI Measurement: A method used to assess plant transpiration?

Answer 6: The possibility of measure relative changes in transpiration using a thermal imager has been demonstrated in a few of papers (http://dx.doi.org/10.3920/978-90-8686-814-8_63, https://doi.org/10.3390/rs13224710, https://doi.org/10.3390/rs11070757). The method does not allow to determine the absolute values of transpiration intensity in mol/m2 s, however, it allows to correctly assess changes in transpiration caused by different stressors. We also used this method earlier in our research (https://doi.org/10.3390/ijms24010491, https://doi.org/10.3390%2Fplants12040826). Additionally, we showed similarities in the dynamics of changes in transpiration intensity and leaf temperature (https://doi.org/10.3390/plants9101364).

Point 7: L498. Was the study conducted on potatoes or tobacco?

Answer 7: Thank you for your comment! Only on potato. We have corrected the error in the text of the manuscript.

Point 8: L510. The region of interest (Figure 8) was located horizontally and at the same level as the track of micrometer light.

Answer 8: Thank you for your comment! The sentence has been corrected.

Point 9:  L519. The scheme of electrode positions on the leaf is shown in Figure 9.

Answer 9: Thank you for your comment! The sentence has been corrected.

Point 10: L538-541. What statistical test was used? I do not see the results of the two-way ANOVA analysis in the results presented. Please cite correctly the software used (Company, country, headquarters).

Answer 10: Thank you for your comment! We added information about the software in Methods. We checked statistical analysis of the data and used multiple t-test. We corrected information about statistical analysis in Methods.

Point 11: Fig. 4. Parts A and B of the graph are not compatible. A. shows control and 200nM NaCl and part B does not include control but shows 100 and 200 nM KCL. In part B - no statistical analysis.

Answer 11: The Figure 4 shows the accumulation of sodium and chlorine ions: for part A, at several time points during the experiment, for part B, the treatment of NaCl, not KCl, is also shown. Part B shows a comparison of sodium accumulation for two NaCl treatment options – 100 and 200 mM. We fixed the figure. We added control values to the part B and we show that compared with the control, there is a significant increase in the sodium concentration in the leaves when treated with both NaCl concentrations, but the comparison between them did not find significant differences.

Point 12: Fig. 5. The graph does not include control plants! There are no statistics.

Answer 12: Thank you for your comment! Here is a comparison between the treatment of NaCl, KCl and sorbitol. For each type of treatment, the difference with the control is shown. We fixed the figure. We have added symbols of the statistical significance of the differences between each treatment variant and the control. We also indicated the statistical significance of the differences between the types of treatment, where it was.

Point 13: Fig. 6. No line for control plants? Statistics show differences from control?

Answer 13: We fixed the figure. We have added lines of control to Figure 6. Yes, there is statistical comparison versus control, and it absented in the original version of the manuscript. We fixed it.

Point 14: Fig. 7. No lines for control plants? Statistics show differences from control? The graph shows the dynamics of change. Were there no statistical differences between the test subjects at 6 and 8 o'clock compared to the control? Perform a factor analysis.

Answer 14: We fixed the figure. We have added lines of control to Figure 7. Statistical differences from the control are shown. There were not significant differences between the different variants of treatment.

Point 15: Fig. 12. I am not sure which part shows La3+ pretreatment and which part shows a lack of La3+. What was the control? There are no control bars on the graph. No statistics.

Answer 15: The figure 12 shows the differences from the control for different agents (NaCl, sorbitol, KCl). We have added information about this to the caption of the figure. We have identified the significancy of the differences versus control. Comparison of groups with lanthanum pretreatment (fill columns) and without (full columns) did not reveal significant differences, and therefore there are not corresponding symbols.

Point 16: Fig. 13. The statistics are not clear on the graph. Were there statistical differences between the tested leaves (1, 2, and 3)?

Answer 16: Thank you for your comment! We added the difference in photosynthesis activity, transpiration and cytosolic pH between the strata to Figure S1 and S2 in Supplementary. We saved information about leaf of strata 2 to avoid an excess of information. Effect of lanthanum pretreatment is the greatest information in Figure 13, we combined Figure 11 and Figure 13.

Point 17: In Fig. 14, parts of the graph A, B, and C are not compatible. They do not show the same objects. No statistics. Were there statistical differences between 6 and 8 o'clock? Did the studied parameters for leaves 1–3 differ statistically?

Answer 17: Thank you for your comment! Parts A, B and C of Figure 13 (previously Figure 14) show changes in cytosolic pH in different parts of the plant, so we believe that they may be compatible. We did not find significant differences in Part A. Significant differences also absented in the leaves of different strata during 100 mM NaCl treatment, not 200 mM NaCl treatment. We presented information about this in Figure S1 and S2 in Supplementary.

Reviewer 2 Report

Comments and Suggestions for Authors

Dear Authors,

Manuscript titled "Salt-induced early changes in photosynthesis activity caused by root-to-shoot signaling" contains valuable research on the impact of salt stress on photosynthetic activity in potato plants. The research involves a large number of measurements.

Below are some comments on the manuscript:

- Abstract - contains too little information regarding the obtained research results. It lacks a sentence summarizing the results obtained

- Introduction - is written too concisely. It is worth supplementing e.g. L: 29-33 and providing more examples of the harmful effects of soil salinity. Please also explain the physiological processes described in more detail. The Introduction also lacks information about the description of the potato, which is the plant under study. There is also a lack of research purpose and research hypothesis

- Results - well described, the charts are of high quality and very interesting

- Discussion - describes in detail the obtained research results based on numerous literature

- Materials and Methods - why was tap water used for testing instead of deionized water?

- Conclusions - is poor and needs to be re-written. There are no specific research results and a summary. There is a lack of practical application of research results and prospects for further experience.

Author Response

Response to Reviewer 2

Dear Reviewer,

We are grateful to you for your comments and suggestions for improving our work. We have revised the manuscript according to your comments. Below are the answers to your questions mentioned in the review (the reviewer's comments are highlighted in bold, our answers follow in plain text).

Point 1: Abstract - contains too little information regarding the obtained research results. It lacks a sentence summarizing the results obtained

Answer 1: Thank you for your comment! We have made changes to the annotation. At the same time, we want to meet the requirements of the Journal, according to which the volume of the abstract should not exceed 200 words.

Point 2: Introduction - is written too concisely. It is worth supplementing e.g. L: 29-33 and providing more examples of the harmful effects of soil salinity. Please also explain the physiological processes described in more detail. The Introduction also lacks information about the description of the potato, which is the plant under study. There is also a lack of research purpose and research hypothesis

Answer 2: Thank you for your comments! We have included in the Introduction information about the importance of potato for agriculture. We have increased the information about the harmful effects of salinity on plants. We have supplemented the article with links to reviews, which set out modern ideas about the harmful effects of salinity, including on potato plants. We also explicitly formulated the purpose of the study.

Point 3: Results - well described, the charts are of high quality and very interesting

Answer 3: Thank you for your positive assessment! We have made some changes to the figures in accordance with the comments of another reviewer. Hopefully, these changes will improve the perception our results.

Point 4: Discussion - describes in detail the obtained research results based on numerous literature

Answer 4: Thank you for your positive assessment!

Point 5: Materials and Methods - why was tap water used for testing instead of deionized water?

Answer 5: Thank you for the comment! We used as a control and for dilution of reagents the same water that we usually use for irrigation and for hydroponics. We assume that the use of this water will not be additional stress for the plant, unlike deionized water.

Point 6: Conclusions - is poor and needs to be re-written. There are no specific research results and a summary. There is a lack of practical application of research results and prospects for further experience.

Answer 6: Thank you for your comment! We have changed the conclusion in the manuscript. In a modified manuscript, the Conclusion contains a common result of the research. We also described promising ways for further research and the importance of work for agriculture.

Round 2

Reviewer 1 Report

Comments and Suggestions for Authors

The manuscript has been reliably revised. I recommend its acceptance in its present form.

I congratulate the authors on their interesting research.

Best Regards,